# Structure of the initiation-competent RNA polymerase I and its implication for transcription

Michael Pilsl[1], Corinne Crucifix[2], Gabor Papai[2], Ferdinand Krupp[2], Robert Steinbauer[1,†], Joachim Griesenbeck[1], Philipp Milkereit[1], Herbert Tschochner[1] & Patrick Schultz[2]

Eukaryotic RNA polymerase I (Pol I) is specialized in rRNA gene transcription synthesizing up to 60% of cellular RNA. High level rRNA production relies on efficient binding of initiation factors to the rRNA gene promoter and recruitment of Pol I complexes containing initiation factor Rrn3. Here, we determine the cryo-EM structure of the Pol I-Rrn3 complex at 7.5 Å resolution, and compare it with Rrn3-free monomeric and dimeric Pol I. We observe that Rrn3 contacts the Pol I A43/A14 stalk and subunits A190 and AC40, that association re-organizes the Rrn3 interaction interface, thereby preventing Pol I dimerization; and Rrn3-bound and monomeric Pol I differ from the dimeric enzyme in cleft opening, and localization of the A12.2 C-terminus in the active centre. Our findings thus support a dual role for Rrn3 in transcription initiation to stabilize a monomeric initiation competent Pol I and to drive pre-initiation complex formation.

[1] Universität Regensburg, Biochemie-Zentrum Regensburg (BZR), Institut für Biochemie, Genetik und Mikrobiologie, Lehrstuhl Biochemie III, 93053 Regensburg, Germany. [2] Department of Integrated Structural Biology, IGBMC (Institut de Génétique et de Biologie Moléculaire et Cellulaire) INSERM, U964; CNRS/Strasbourg University, UMR7104 1, rue Laurent Fries, BP10142, 67404 Illkirch, France. † Present address: Sandoz GmbH, Biochemiestraße 10, 6250 Kundl, Austria. Correspondence and requests for materials should be addressed to H.T. (email: herbert.tschochner@ur.de) or to P.S. (email: patrick.schultz@igbmc.fr).

Although all of the nuclear RNA polymerases share common features in composition and basic transcriptional mechanisms[1,2], they are highly specialized to recognize and specifically transcribe their target genes[3,4]. In most eukaryotes RNA polymerase I (Pol I) recognizes only one promoter and synthesizes a large precursor transcript which is processed into the mature 5.8S, 18S and 25/28S ribosomal RNAs (rRNAs). Initiation at the Pol I promoter requires several basal transcription initiation factors. In the yeast *S. cerevisiae* these are the TATA-binding protein (TBP), and the Pol I-specific factors as Rrn3 (ref. 5), the core factor (CF) which contains the three subunits Rrn6, Rrn7 and Rrn11 (refs 6,7), and the upstream activating factor (UAF) consisting of Rrn5, Rrn9, Rrn10, the histones H3 and H4 and UAF30 (refs 8,9). CF, TBP and UAF provide a promoter-bound platform to which a complex of Pol I with Rrn3 is recruited[5,10]. Several aspects of Pol I transcription initiation are conserved between yeast and mammals. Conservation in sequence and function is observed for TBP, Rrn3/TIFIA and Pol I (ref. 11,12) (reviewed in ref. 1). Mammalian SL1 and yeast CF share some functional and structural properties[6,13,14]. Importantly, complex formation of Pol I with Rrn3 was identified as a regulated key step in yeast and mammalian Pol I transcription initiation (reviewed in ref. 15). In yeast, cellular extracts <55% of Rrn3 and <5% of Pol I were estimated to be incorporated in the salt-stable initiation competent Pol I-Rrn3 complex[10,16,17]. This suggested that only a subpopulation of Pol I is compatible with complex formation. When Pol I switches from initiation to elongation, the Pol I-Rrn3 complex is disrupted[10,16,18,19]. The Pol I subunit A43 was shown to be crucial for Pol I-Rrn3 complex formation. Mutations in the central part of A43 abolished Pol I-Rrn3 complex formation, resulted in a temperature sensitive growth phenotype and fractions containing mutated Pol I failed in transcription initiation[20]. Furthermore, the non-essential subunit A49 contributes to the recruitment of Pol I-Rrn3 to the promoter and plays a role in Rrn3 release from the polymerase after promoter clearance[19]. Although there is evidence that the formation of the Pol I-Rrn3 complex might depend on posttranslational covalent modifications *in vivo*[21,22], the molecular requirements for Rrn3-Pol I interaction remain to be defined. Previously, a model of the interaction interface between the transcription factor and the polymerase has been proposed[22,23]. This model was based on the crystal structure of Rrn3 (ref. 22), on a Pol I homology model[24] and on two BS3 (bis-sulfosuccinimidyl-suberate) crosslinks between Rrn3 Lysine K558 and Lysines K582 and K329 of the Pol I subunits A190 and AC40, respectively. Thus, it was suggested that the serine patch of Rrn3 is oriented towards the Pol I surface, and Rrn3 stretches from the RNA exit tunnel down to subunits AC40/19, while contacting subunits A43/14 with its central part[22].

Recently, the crystal structure from Pol I dimers provided insights in the architecture of the enzyme at atomic resolution[25,26]. Interestingly, direct comparison with the Pol II structure showed several differences, some of which are probably a consequence of Pol I dimerization. The C-terminal part of A43, 'the connector' domain, invades between the clamp and the 'protrusion' domains of the neighbouring Pol I molecule. It is possible that this stable interaction contributes to significant widening of the nucleic acid binding 'cleft', when compared with the corresponding structure in Pol II. The wider cleft accommodates an extended Pol I specific 'expander' loop, which mimics the DNA backbone. The absence of the expander loop in the opened cleft of some (dimeric) crystal structures[26,27] suggests that the expander loop is a mobile domain. Other possible consequences of the widened cleft could be that the 'bridge helix' in the active centre is unwound[26], and that the

C-terminal domain of subunit A12.2 is inserted into the nucleotide triphosphate entry pore. The C-terminal domain of A12.2 is homologous to the C-terminus of TFIIS adopting a similar structure in the active centre of a Pol II-TFIIS complex, which is transcriptionally stalled after backtracking[28–30]. The inserted C-terminus of TFIIS stimulates RNA cleavage to resume Pol II-dependent RNA chain elongation.

The transition from a dimeric and expanded inactive conformation into a more contracted, initiation-active monomeric conformation was suggested to play a role in Pol I transcription regulation[25]. In this model, Pol I is in an equilibrium of dimers and monomers. Monomer formation requires the reorientation of the connector. Monomers are thought to be inactive until the release of the expander, perhaps due to DNA loading and cleft contraction. Monomerization also provides the interface for association with Rrn3 and Rrn7 (ref. 25).

In summary, the high-resolution structures of the Pol I enzyme[25,26] and Rrn3 (ref. 22) provided important insights into the molecular architecture of the rRNA synthesizing machinery. Here, the structure of the yeast Pol I-Rrn3 complex is solved by single-particle electron cryo-microscopy at 7.5 Å resolution, and strongly support the previously suggested interaction interface between Rrn3 and Pol I (ref. 22). Additionally, several differences to the Pol I crystal structures are detected, which correlate with the initiation competence of the monomeric complex. These include changes in the active centre of Pol I like the re-organization of the expander, and the displacement of the A12.2 C-terminal domain together with closing of the cleft. Furthermore, cryo-electron microscopy (cryo-EM) detects a not fully assigned density over the cleft, which may correspond to the domain of A49 not resolved in the high-resolution structures. The position of the presumed A49 domain is in good agreement with earlier findings[31,32]. The use of a minimal promoter-dependent *in vitro* transcription system supports a role for the tandem winged-helix (tWH) domain of A49 in transcription initiation and elongation in line with its suggested functional roles[19,24,33].

## Results

**Purified initiation-competent yeast Pol I-Rrn3 complex.** Initiation of Pol I transcription in yeast cell extracts depends on the formation of a salt resistant complex between Pol I and the initiation factor Rrn3 which was previously described as initiation competent Pol I (ref. 10). In yeast whole-cell extracts, the Pol I-Rrn3 complexes represent only a minor population of total cellular Pol I. We have recently described a purification protocol to obtain highly enriched Pol I-Rrn3 complex, which, together with recombinant CF, was active in a minimal promoter-dependent transcription system[34]. The protocol for isolation of the stable Pol I-Rrn3 complex could be further improved (Methods section), yielding a complex consisting of almost stoichiometric amounts of Pol I subunits and Rrn3 (Fig. 1a, first panel). A similarly prepared yeast extract from a different strain served as source to purify a Pol I complex, which was largely devoid of Rrn3 (see Methods section, Fig. 1a, second panel). Incubation of Rrn3-free Pol I with over-stoichiometric amounts of purified recombinant Rrn3 and CF (Fig. 1a, last panel) resulted in promoter-dependent transcription, whereas no transcript was detected in reactions lacking either one of the components (Supplementary Fig. 1; see also Fig. 6b, lanes 13–15). The optimal amount of Rrn3 at a given concentration of Pol I and CF was determined in titration experiments (Supplementary Fig. 2). Comparison of the transcriptional activity of the two Pol I-containing fractions showed clear differences when the same concentration of Pol I and Pol I-Rrn3 and optimized amounts of

CF and Rrn3 were used (Fig. 1b). Whereas Rrn3-free Pol I produced higher amounts of RNA in promoter-independent transcription from tailed templates, the Pol I-Rrn3 complex was up to 10-fold more active in promoter-dependent transcription (Fig. 1b). This result suggested that complex formation between Rrn3-free Pol I and recombinant Rrn3 *in vitro* was poor under the conditions used in this study. The successful purification of the initiation competent Pol I-Rrn3 complex from yeast extracts encouraged us to study its three-dimensional (3D)-structure using cryo-EM.

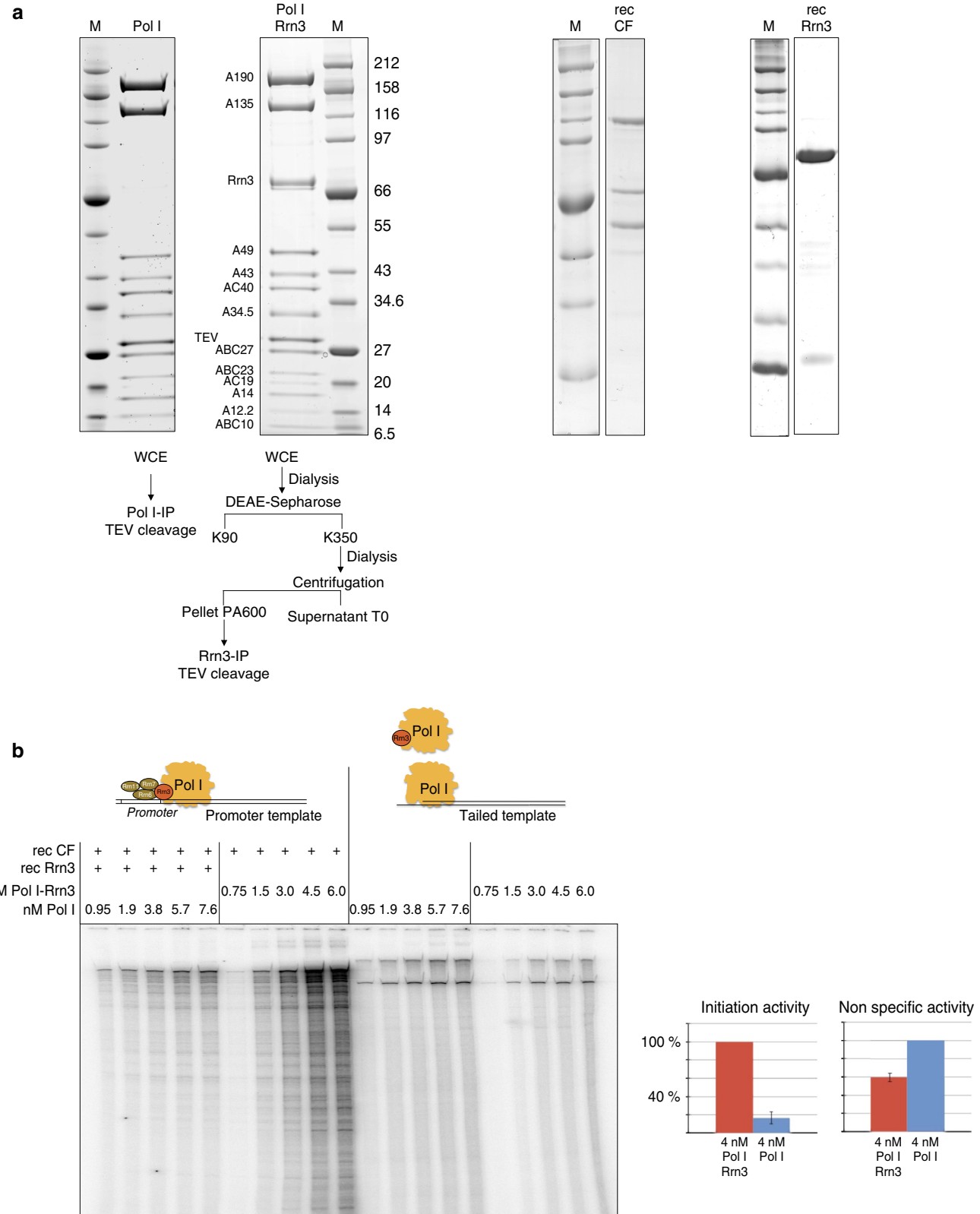

**Cryo-EM structure of the Pol I-Rrn3 complex**. A total of 49,583 molecular images of the frozen hydrated, glutaraldehyde cross-linked Pol I-complexes were collected and sorted to separate free Pol I from intact complexes (32,438). The refined 3D model of the Pol I-Rrn3 complex was solved at an overall resolution of 7.5 Å. This allowed to visualize secondary structure elements of the enzyme and its associated cofactor and enabled precise docking of their independently determined atomic structures into the cryo-EM map[22,25,26] (Fig. 2a, Supplementary Fig. 3). The elongated Rrn3 molecule interacts through its N-terminus with the A43/A14 'stalk', contacts the 'dock' region of subunit A190 and reaches the AC40 and AC19 subunits with its C-terminal end as suggested by cross-linking and homology modelling data[22,23].

While the position of the A14 subunit is not affected when its residues 83–85 interacts with residue 224 of Rrn3, the A43 subunit is reorganized upon interaction (Fig. 2b). The bundle of beta strands forming the oligonucleotide/oligosaccharide-binding 'OB' fold (residues 127–251) and the N-terminal 'tip' domain of

A43 are not affected, while residues 274–316 of A43 are not detected and are probably re-positioned on Rrn3 binding (Supplementary Fig. 4). These residues encode the connector helix which is an essential determinant of Pol I dimerization[25–27], suggesting that the binding of Rrn3 interferes with dimer formation. Finally, the Rrn3 serine-patch identified as important for Pol I binding[22] is involved in the interface.

The helix-forming residues 243–251 of Rrn3 contact the 'dock' domain of the largest A190 subunit and particularly helix 549–564 and residues 564–573 which are part of the Pol I-specific region α12a (Fig. 2c). Finally, the end of helix-forming residues 554–542 of Rrn3 are in close contact with C-terminal and N-terminal loops of AC40 (residues 334) and AC19 (residues 44–49), respectively (Fig. 2d). This is consistent with cross-linking data using BS3 (bis-sulfosuccinimidyl-suberate), which identified with high confidence a contact between lysine 558 of Rrn3 and AC40 lysine 329 and A190-lysine 582, respectively[22] (Supplementary Fig. 5).

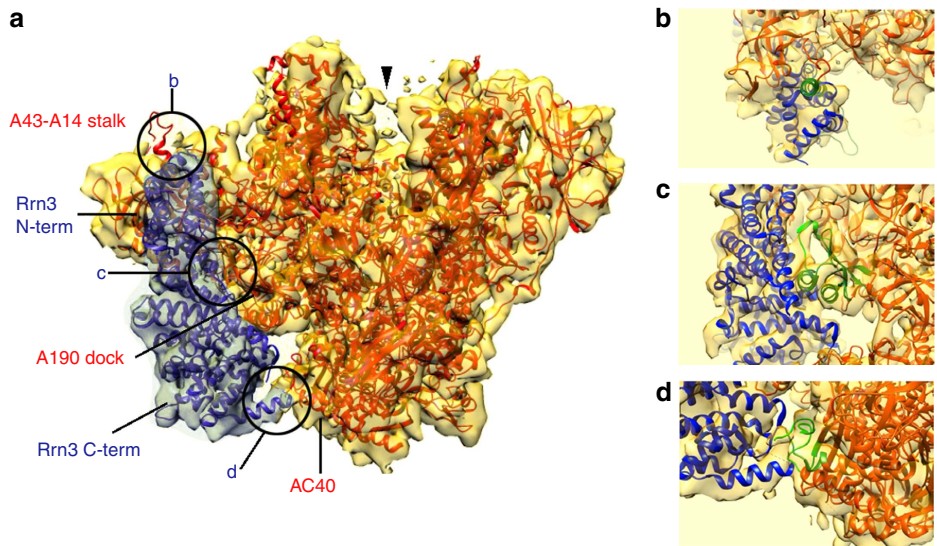

**Figure 2 | Cryo-EM structure of the Pol I-Rrn3 complex.** (**a**) Atomic structure of Pol I (red, PDB accession numbers 4C2M (ref. 25) and 4C3H (ref. 26)) and of Rrn3 (blue, PDB accession number3TJ1 (ref. 22)) docked into the cryo-EM structure of the Pol I-Rrn3 complex (transparent yellow envelope). Regions enlarged in **b**–**d** are highlighted. Arrow head shows the weak density bridging the cleft poorly visible at this threshold. (**b**) Close-up view of the Pol I-Rrn3 complex showing the interaction of the N-terminus of Rrn3 with the A43-A14 stalk. The C-terminal part of A43 (residues 273–316, shown in green) is displaced on Rrn3 binding. (**c**) Close-up view of the Pol I-Rrn3 complex on the central domain of rrn3 interacting with the dock domain of the largest Pol I subunit A190 containing the Pol I-specific region α12a (green). (**d**) Enlarged view of the Pol I-Rrn3 complex showing the interaction of the C-terminus of Rrn3 (blue) with the N- and C-terminal loops of AC40 (green).

**Figure 1 | Purified Pol I-Rrn3 complex from yeast whole-cell extracts is more active in transcription initiation than purified Pol I complemented with bacterially expressed Rrn3.** (**a**) Protein fractions used for either transcription initiation in a minimal promoter-dependent transcription assay or in non-specific (tailed template) transcription. Proteins were separated by SDS–PAGE using a 4–12% gradient gel which was stained with SimplyBlue SafeStain (ThermoFisher). Pol I was affinity purified from strain BY4741 A135-TEV-ProtA (y2423) using a ProtA-TEV-tag fused to subunit A135 (see Materials and Methods). Purified Pol I (10 μg) were loaded. Molecular weight standards (M) and Pol I subunits which were identified by mass spectrometry are indicated. Pol I-Rrn3 complex was derived from yeast strain y2183 (BSY420-Rrn3-TEV-ProtA-His7-Tag) which was transformed with plasmid 729 (Ycplac111-GAL-Rrn3-TEV-ProtA-HIS). This strain overexpresses Rrn3 in the presence of galactose. The complex was affinity purified from the initiation-active fraction PA600 (ref. 17) using a ProtA-TEV-tag fused to Rrn3 (Materials and methods section). Pol I-Rrn3 complex (6 μg) were loaded. Recombinant Rrn3 (rec Rrn3) and CF (rec CF) were expressed in *E. coli* and purified according previously published protocols[22,48] with some modifications (Materials and methods section). MonoQ elution fraction (10 μl) of Rrn3 and 11 μl of a Superose 6 elution fraction of CF respectively, were separated on a 10% SDS-gel and Coomassie stained. (**b**) Comparison of differently derived Pol I fractions in promoter-dependent and non-specific transcription assays (10 nM template concentration, end point labelling after 30 min). Rising concentrations of Pol I or Pol I-Rrn3 complex (concentrations in the assays are indicated) were assayed together either with recombinant His6-Rrn3 (70 nM) and CF (20 nM) or CF (20 nM) in promoter-dependent transcription (left panel). The same concentrations of Pol I and Pol I-Rrn3 were used in non-specific transcription assays (middle panel). A quantitative comparison of the two different Pol I fractions (4 nM) in the two transcription assays is indicated on the right.

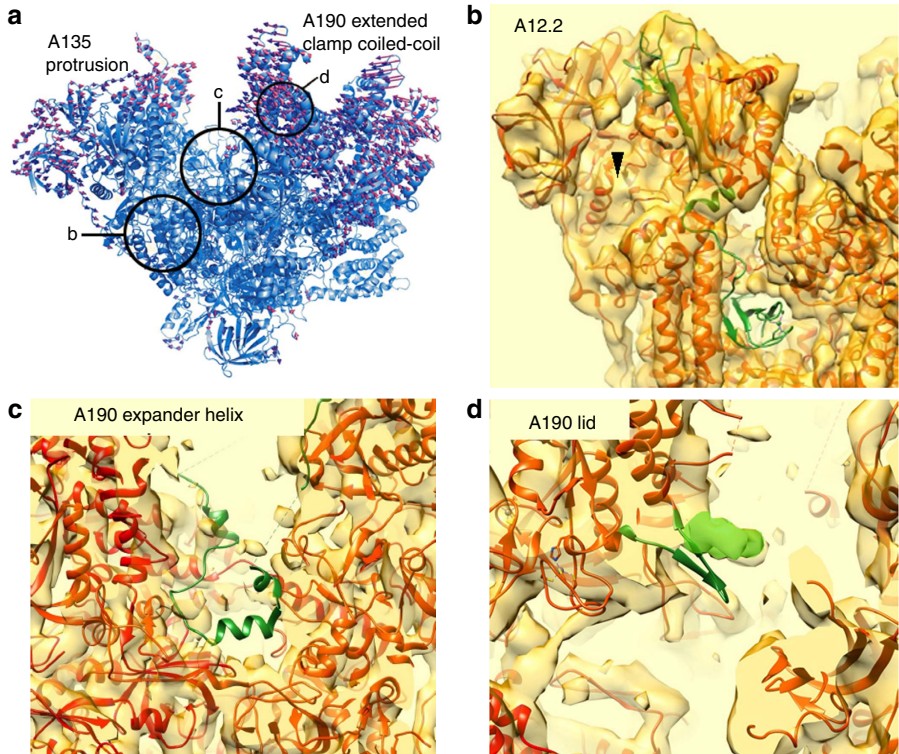

**Figure 3 | Structural changes of Pol I in the Rrn3-bound conformation as compared with the crystal form.** (**a**) Flexible fitting of the atomic structure of Pol I into the cryo-EM structure of the Pol I- Rrn3 complex. The displacement vectors (red) associated to each residue when moving from the crystal structure (blue) to the cryo-EM model show the closing of the cleft and the upward movement of the A43-A14 stalk. Structural regions enlarged in **b**–**d** are highlighted by circles. (**b**) Crystal structure of the A12.2 subunit (green) whose C-terminal TFIIS-like domain is not located in the pore of the Pol I-Rrn3 cryo-EM structure. (**c**) Crystal structure of the expander helix (green) of the A190 subunit (residues 1,361–1,378) which is not resolved in the cryo-EM structure. (**d**) Position of the lid loop of subunit A190 (residues 368–380) in the crystal structure (green ribbons) as compared with the cryo-EM structure (green envelope).

**Fitting Pol I into the Pol I-Rrn3 cryo-EM model.** Interestingly, significant changes were observed in the structure of Pol I. The active centre cleft is more contracted in the cryo-EM map than in the crystal structure. The displacement vector field obtained by comparing the crystal form with the normal mode fitted version shows that the 'clamp' domain and the A43-A14 stalk moved inwards (Fig. 3a). The two long helices that form the clamp core domain of A190 are displaced by 8 Å. (Supplementary Fig. 4a). On the other side of the DNA-binding cleft an inward movement of 3.5 Å is also measured at the tip of the 'protrusion' domain of A135, indicating that the cleft closes by 11.5 Å when compared with the crystal structures. In addition, the C-terminal part of A12.2, which holds the TFIIS homology region, is absent in the cryo-EM map while it is positioned in the pore in the crystal structure (Fig. 3b). Interestingly, A12.2 is not dissociated from Pol I since the N-terminal part, homologous to Rbp9, including two helices is perfectly resolved and is located similar to its position in the crystal structure (Fig. 3b). This observation indicated that the C-terminal domain of A12.2 dissociates from the active site and is probably flexible since no similar density was detected in the map. Another important difference to the crystal structure is found within the active site. While all helices are resolved, no density is observed for the Pol I-specific expander domain and the expander helix, an element present in the active site in several crystal structures (Fig. 3c)[25,26]. The fact that the expander helix/extended loop is missing in some crystal forms[26,27] and could not be traced in the electron density map suggests that it is flexible. Finally, we observed that the A190 'lid loop' (residues 368–380) is perfectly resolved in the EM map, but its position within the RNA exit

channel is slightly shifted (Fig. 3d), suggesting that the RNA exit channel is less occluded than in the Pol I crystal structure.

When the threshold is lowered, a weak density can be detected over the cleft bridging the clamp to the 'protrusion' domain, suggesting that this bridge is present in only a subpopulation of Pol I molecules (arrow head in Fig. 2a). The position of this bridge is consistent with the proposed location of the mobile C-terminal tWH domain of A49 determined by chemical cross-linking and mass spectrometry[32] and could correspond to the A49 position determined previously by immuno-EM (ref. 31).

**Rrn3-free Pol I monomers differ from Pol I dimers.** To ascertain the exact contribution of Rrn3 to the observed changes in Pol I structure, we analysed the structure of purified Rrn3-free Pol I, which was found in a monomeric form for which high-resolution information was not available and as dimers as in the crystal structures. In solution, the two forms co-exist and the monomers (108.214 particles 28%) were separated from the dimers (141,024 particles 72%) *in silico*. At a resolution of 7.5 Å, the isolated Pol I monomer structure was very similar to the Rrn3-bound enzyme (Supplementary Fig. 6). In particular, the C-terminal domain of A12.2 is not detected, whereas it is present in the centre of the dimer (Fig. 4a,d,g; Supplementary Fig. 7) and the position of the clamp is almost identical in both structures indicating that the cleft has the same width. The lid loop was positioned slightly differently than in the crystal structure but still in a way that it partly occludes the RNA exit channel (Fig. 4c,f,i). The expander helix was clearly not in the same position as in the

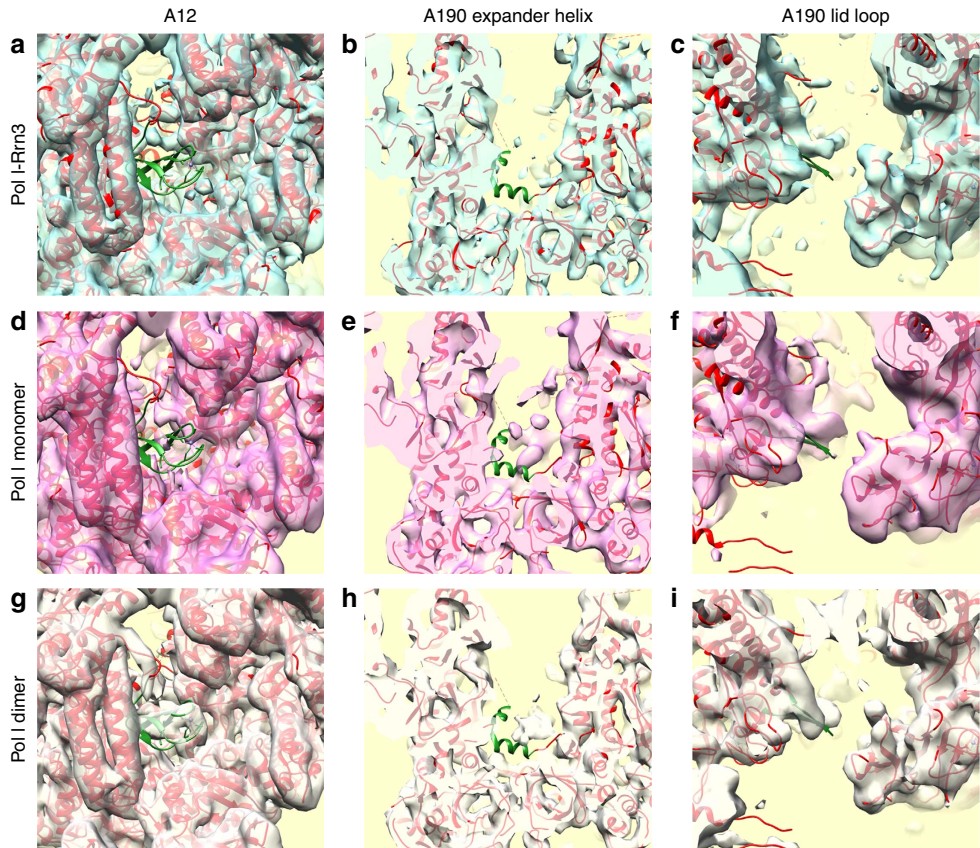

**Figure 4 | Comparison of key structural features in different Pol I conformational states.** The position of the A12.2 C-terminus (**a,d,g**) the A190 expander helix (**b,e,h**) and the A190 lid loop (**c,f,i**) are shown in the cryo-EM maps of the Pol I- Rrn3 complex (**a–c**) the Pol I monomer (**d–f**) and the Pol I dimer (**g–i**).

early crystal structures but a new density is placed in the active site which could correspond to a different position of the expander domain (Fig. 4b,e,h). The monomeric Pol I form seems to be slightly depleted in A49/34.5 since the corresponding density is weaker than in the Pol I-Rrn3 complex, but the bridge over the cleft is detected as a faint electron density (data not shown). Altogether, this comparison indicated that the major conformational changes in Pol I are not specific for the Rrn3-bound complex, but rather properties of monomeric Pol I.

**Similar cryo-EM and crystal structures of Pol I dimers**. The analysis of the dimeric Pol I form by cryo-EM was important to understand the differences with the crystal structures (Supplementary Fig. 8). A final resolution of 7.8 Å was obtained for the entire dimer, which shows a slight movement between the two monomers. This could be partially corrected by analysing a single monomer thus reaching a resolution of 6.8 Å. The A43-A14 stalk is essential for dimerization and the stalk of one monomer interacts with the DNA-binding cleft of the second monomer. The connector helix in A43 plays a crucial role for dimerization and contacts the 'protrusion' domain of A135 close to the Pol I-specific insertion α11a. Whereas, the A43 connector helix is poorly resolved in the monomeric form of Pol I, and is displaced on binding of Rrn3, it is perfectly resolved in the dimer at the position determined by X-ray crystallography (Supplementary Figs 6 and 8). Moreover, the cryo-EM dimer map is comparable to the crystal structure with regard to cleft opening, and for the density corresponding to the C-terminal domain of A12.2, which is clearly detected at the same position (Fig. 4g). However, the

expander helix observed in some of the crystal structures, is not in the same position, while a new density appears in the active site as seen for the isolated monomer (Fig. 4h). Interestingly, in all three structures the catalytically important 'bridge' helix appeared to be partially unwound in its central part, indicating that even when the expander helix is absent the bridge helix does not fold properly (Supplementary Fig. 9). The A190 lid loop is well resolved in the cryo-EM dimer map and it adopts the same position than in the crystal form (Fig. 4i). Furthermore, the density bridging the DNA-binding cleft is weakly detected in all maps suggesting that the corresponding protein domain is not stably positioned.

**The C-terminal domain of A49 spans the Pol I cleft**. To better define the spurious density detected in all the cryo-EM maps above the DNA-binding cleft, we analysed the density variations within the dimer-forming monomers by 3D classification. This analysis revealed a protein density, which can adopt different positions. The most abundant conformation, representing ∼35% of the molecules, depicts this domain more clearly (Fig. 5a) although residual flexibility hindered resolving secondary structure elements. This additional density occupies a volume of ∼17 kDa, which is consistent with the mass of the tandem winged domain (tWD) of A49. This domain was not resolved in the crystal structures and has been shown to cross-link to both sides of the cleft[32]. The density bridging the cleft contacts several Pol I-specific sequences not found in Pol II: the two regions flanking the A190 expander domain (residues 1,320–1,337 and 1,440–1,456), the A190 clamp head domain (residues 56–303)

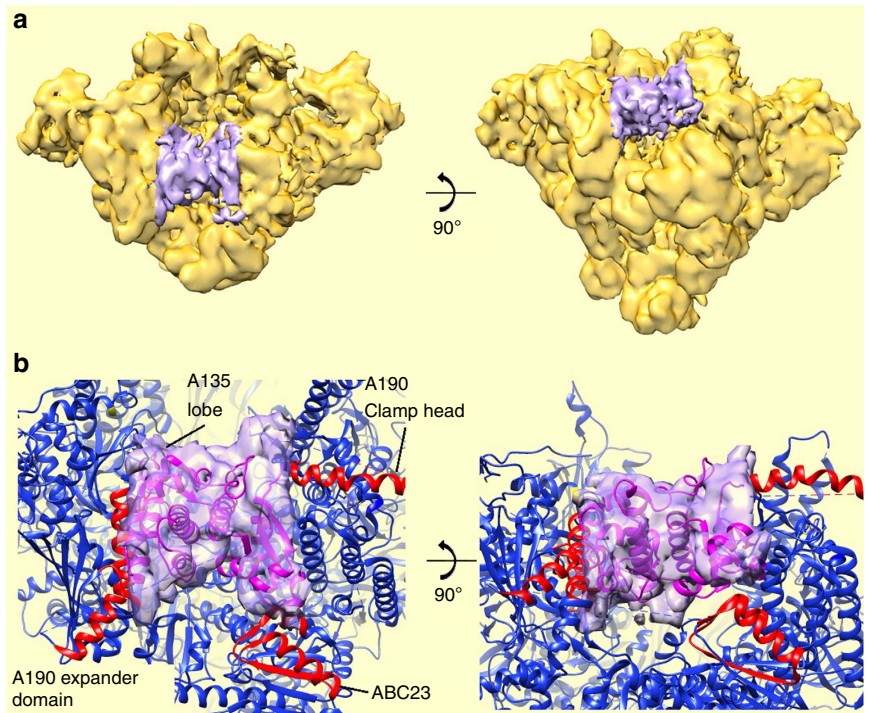

**Figure 5 | The A49 tWD bridges the DNA-binding cleft.** (**a**) Protein density (purple) found in a subpopulation of Pol I molecules. (**b**) Proposed fit of the A49 tWD (PBD accession number 3NFI (ref. 33)) within this density. Residues in contact with the A49 tWD are highlighted in red and correspond to the A190 clamp head domain (residues 256–303), the Pol I-specific regions flanking the A190 expander domain (1,320–1,337 and 1,440–1,456), a Pol I-specific region in the A135 lobe domain (residues 222–226) and the ABC23 subunit (residues 87–123).

and a small loop in the A135 lobe domain (residues 222–226; Fig. 5b). The density also contacted two helices in subunit ABC27 (residues 87–123). The atomic structure of the A49 tWD domain could be fitted in size and shape into this additional density thus confirming that it likely corresponded to the C-terminus of A49 (Fig. 5b). The lower resolution of this part of the cryo-EM map did not provide sufficient secondary structure information to confirm the fitting. In this position, the set of distance constraints obtained from cross-linking data were minimized and 50% of the observed cross-linked lysines were within 30 Å (ref. 32). However, for the other half of the reported intersubunit crosslinks, the concerned lysines are further apart, an observation that probably reflected the flexibility of this domain.

**The C-terminal domain of A49 supports initiation *in vitro*.** The localization of the presumed A49 C-terminus would be compatible with previous data[31,32] which showed its involvement in DNA binding and transcription elongation but also with the suggested TFIIE-like function in initiation of Pol I transcription. Therefore, we re-analysed the role of the C-terminal domain in the promoter-dependent *in vitro* transcription system (Fig. 6). Pol I lacking subunits A49/34.5 (Pol I Δ49) was almost inactive when compared with wild-type (WT) Pol I in promoter-dependent transcription (Fig. 6a, compare lane 6 with lanes 7 and 8). Addition of recombinant A49/34.5 dimer strongly stimulated transcription. In contrast, Pol I Δ49 was more potent in tailed template transcription, while the addition of A49/34.5 dimer showed only moderate stimulation in this assay. This is in line with *in vitro* data that the A49/34.5 dimer supports Pol I processivity[24,33] and with *in vivo* data for a role in transcription initiation and elongation[19]. Consistent with previous analyses[19,33], addition of the A49 tWH domain alone to Pol I Δ49 was sufficient to restore promoter-dependent and tailed

template transcription to a similar level as observed for the addition of recombinant A49/34.5 dimer (Fig. 6b), whereas addition of the A49/34.5 dimerization module had no effect. These results further point to a role of the A49 tWH domain for both, Pol I transcription initiation and processivity.

Altogether, these results showed that the monomeric Pol I structure derived from cryo-EM analyses did not significantly differ from the Pol I-Rrn3 complex. This underlined that Rrn3 association was not required for cleft closing and removal of both the expander helix and the A12.2 C-terminus from the active centre. In contrast to free Pol I molecules, Pol I-Rrn3 complexes existed only as monomers consistent with the observation that Rrn3 binding interfered with Pol I dimerization mediated by A43. To find out whether Rrn3 incubation triggered monomerization of dimeric Pol I, increasing amounts of Rrn3 were incubated with a fraction containing both monomeric and dimeric Pol I (Supplementary Table 4). In comparison to control reactions without Rrn3, only a subtle shift in the populations towards more Pol I monomers could be observed. The ratio of monomers and dimers did not significantly change even after an incubation time for 2 h with a 10-fold excess of Rrn3. This suggested that only a minor subpopulation of Pol I associates with Rrn3, which is supported by previous published studies[10,16] and the result of *in vitro* transcription experiments presented in the present study (see Fig. 1b and Supplementary Fig. 2). Overall, our finding supported previous studies, suggesting that structural, yet unknown features of Pol I, other factors or experimental conditions were required for dimer dissociation and Rrn3-binding[17,21].

## Discussion
The general structural arrangement of the Pol I-specific initiation factor Rrn3 on the Pol I core analysed in this study was as

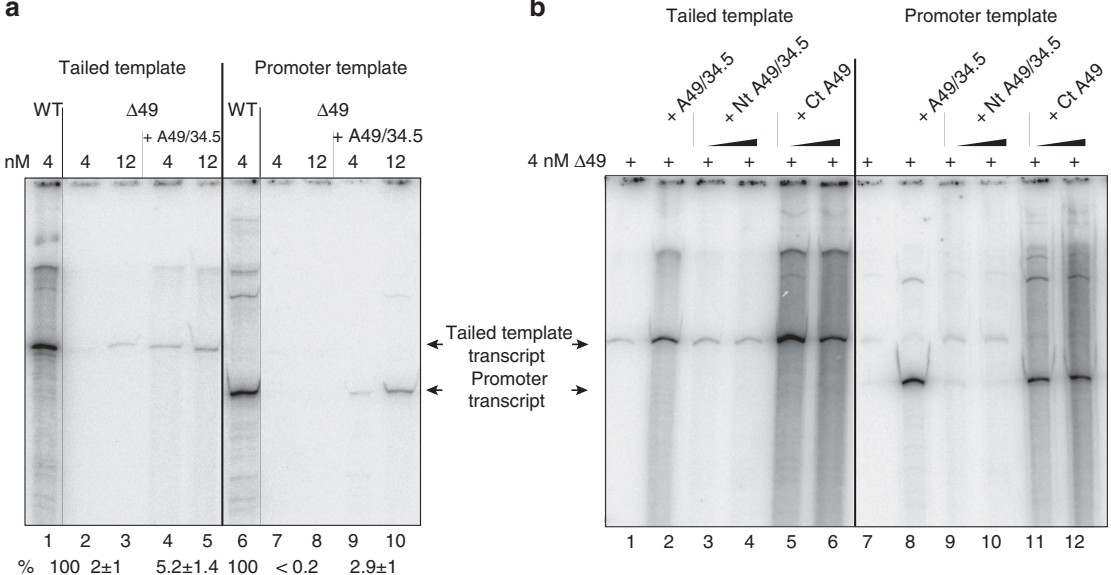

**Figure 6 | The A49/34.5 dimer and the C-terminal part of subunit A49 are required for transcription initiation and support non-specific transcription.**
(**a**) Recombinant subunit A49/34.5 can complement for the transcription defect of Pol I ΔA49. Tailed template assays and promoter specific transcription were performed using 4 or 12 nM WT Pol I or ΔA49 Pol I in the presence and absence of purified recombinant A49/34.5 heterodimer. Promoter-dependent assays were performed in the presence of 70 nM Rrn3 and 20 nM CF. Note that radiolabelled transcripts can be clearly seen in tailed template transcription using Pol IΔA49 after longer exposure, whereas no transcripts can be detected in promoter-dependent assays. (**b**) The C-terminus of A49 is sufficient to complement for initiation in a minimal transcription system which is dependent on recombinant CF, Rrn3 and purified Pol I ΔA49. Same experimental outline as in **a**, but transcription was analysed in the presence of the purified C-terminal domain of A49 (residues 111–426) or the coexpressed N-terminal domain of A49 (1–181) and A34.5 (Materials and methods section).

predicted from cross-linking approaches and modelling of the Rrn3 crystal structure into the homology Pol I/Pol II model[22,23]. Accordingly, Rrn3 is oriented towards subunit A43 making contacts via the Rrn3 serine patch, and stretches along subunits A190 and A135 down to subunit AC40/19 (ref. 22). The orientation of Rrn3 towards the OB fold of subunit A43 is in accordance with previous genetic, biochemical as well as with previous EM data[20]. In this position, Rrn3 can bridge between Pol I and the C-terminus of the CF subunit Rrn6 as it was previously suggested[23]. Another consequence of Rrn3 binding is the re-organization of the A43 C-terminus and the interference with Pol I dimerization by displacing the Pol I–Pol I interaction through the connector helix. Dimer dissociation is important for the structural rearrangements yielding initiation competent Pol I. Interestingly, Rrn3 resembles the Mediator of Pol II transcription head domain in its structure as well as in its interaction with the respective polymerase stalk[22,35,36]. Thus, the comparative structural analysis of Pol I and Pol II may reveal common principles in pre-initiation structure and formation.

The crystal structures of Pol I and Pol II revealed some clear differences, which can be due to the specialization of the enzymes for transcription or an artefact of the particular requirements for crystallization. The most significant differences were that Pol I has an ~10 Å more open cleft, an extended loop (expander) in the cleft, which might mimic DNA and an active centre, which resembles a reactivated backtracked polymerase[28,30]. Furthermore, Pol I crystallized as a dimer in which the C-terminal part of subunit A43 (expander helix) reaches into the cleft of a neighbouring Pol I molecule. Many of these features suggested that Pol I has to undergo conformational changes to initiate transcription. The determination of the cryo-EM structures for Pol I-Rrn3 (and Rrn3-free Pol I) is one further step to understand the formation of the Pol I pre-initiation complex.

The cryo-EM structure of the dimeric form of Pol I strongly resembles the Pol I crystal structure. The structures of Rrn3-Pol I and the Rrn3-free monomeric form of Pol I, in contrast, are more compatible with a transcriptional active enzyme. The cleft is closed by about 11 Å and neither the expander loop, nor the A12.2 C-terminus are found in the active centre. The position of the expander loop seen in some of the crystal structures may hamper interaction with the template. In contrast to Pol II, purified Pol I needs over-stoichiometric amounts of tailed template for efficient RNA synthesis *in vitro*[25]. This might point to the possibility that excess template is required to displace the expander loop (and the C-terminal domain of A12.2 (see below)) from the active centre, thereby converting inactive Pol I into a more transcription competent Pol II-like conformation. Our data suggest that Pol I may adopt a conformation, which is likely compatible with transcription even in the absence of DNA. The interaction between Pol I and Rrn3 could either stabilize or induce such structural changes (see also discussion below).

In the Pol I crystal structure the TFIIS-like C-terminal domain of A12.2 occupied the active centre in the crystal structure. Its position is very similar to that of TFIIS in Pol II after backtracking when TFIIS stimulates intrinsic Pol I cleavage of the 3′ RNA extension[28,30]. Our cryo-EM analyses revealed that, in analogy to the homologous Pol III subunit C11 (ref. 37) and the Pol II factor TFIIS (refs 28,30), the C-terminal domain of A12.2 is not an integral part of the active centre, and that it can be removed from its position in the nucleotide entry pore. However, in contrast to Pol II, and similar to C11, the RNA cleavage-supporting polypeptide chain is tightly associated with Pol I as another example for 'built-in transcription factors' in the enzyme[24,33]. Similar to TFIIS this might help to remove stalled transcription complexes more efficiently[38], thus increasing processivity[39,40]. Consistently, Pol I recovers mostly by RNA cleavage for backtracks larger than 3 nt, whereas Pol II without

TFIIS uses also 1-D diffusion to regain transcription elongation. The 'built-in' cleavage activity may prevent frequently occurring transcriptional arrests which would be especially disadvantagous for the highly transcribed rRNA genes of fast dividing cells[41].

Another dynamic module, which appears to be involved in transcription initiation, is the C-terminal tWH domain of subunit A49. This domain structurally resembles the Pol II transcription initiation factor TFIIE (ref. 33). Ectopic expression of the tWH domain was sufficient to suppress impaired Rrn3 and Pol I recruitment to the rRNA gene in A49 deletion strains[19]. Our *in vitro* data are compatible with a role of the tWH domain in rDNA transcription initiation, although it also enhanced transcript elongation from a tailed template. The position of the tWH domain was not resolved in the Pol I crystal structure, but the cryo-EM data in this study together with previous immuno-EM (ref. 31), and cross-linking analyses[23,32] indicate that it forms a bridge over the DNA-binding cleft. The structure of the entire A49/34.5 heterodimer strongly resembles the structures of the Pol II transcription factors TFIIF and TFIIE, and indicates that they may have similar functions[33]. The cryo-EM data are in line with the assumption that the tWH domain is mobile and likely needs to be displaced to allow access of the promoter DNA to the Pol I active site.

The interesting concept of 'built-in transcription factors' appears to be a general feature for yeast Pol I and Pol III systems (reviewed in ref. 2). Thus, instead of relying on limiting factors such as TFIIS, TFIIE or TFIIF functional counterparts have been incorporated in the enzymes in form of stably associated *bona fide* polymerase subunits (for example, A12.2, A34.5 and A49 in case of Pol I). This feature might be the reason for the extraordinary efficient transcription initiation of RNA polymerases I and III.

One remaining significant difference within the active centre of Pol II when compared with the active centres of the three different cryo-EM structures of Pol I (Pol I dimer, Pol I monomer and Pol I-Rrn3) is the conformation of the bridge helix. For Pol II a mechanism was proposed in which a switch between a partially unfolded and a completely folded bridge helix and the resulting bending is important for DNA, and the DNA/RNA hybrid translocation[42,43]. For Pol I it was proposed that the partially unfolded bridge helix is a consequence of the significantly wider cleft[26]. Thus, it was predicted, that cleft closing might induce complete folding of the bridge helix and opening of the RNA exit channel with concomitant inside movement of A135 domains[26]. This structural rearrangement would be necessary for anchoring of the transcription bubble. Whereas the latter two transitions can be seen in the initiation competent Pol I-Rrn3 complex and the Rrn3-free Pol I monomers, the bridge helix remained partially unfolded in all cryo-EM structures. It is possible that DNA binding is required for the complete folding, or that the unfolded bridge helix is a Pol I-specific feature. In fact, exchange of two amino acids, which change the amino acid sequence of Pol I bridge helix into the amino acid sequence of the Pol II helix, led to alterations of transcription speed and processivity in the respective Pol I mutant (M.P. and H.T. unpublished observations). This indicated that the Pol I-specific bridge helix is important for proper Pol I activity.

Currently it is not known whether different subpopulations of Pol I monomers and dimers exist *in vivo*. It is, however, tempting to speculate that these two different forms of Pol I are involved in Pol I transcription regulation as it was previously suggested[17,20,25]. It is possible that Pol I dimers are a stable storage pool for the enzyme, which is kept in an inactive state. In response to changes in physiological situations, the pool of Pol I dimers could be quickly activated into monomers to adjust cellular ribosome biosynthesis. Regulatory processes like phosphorylation, DNA-association or binding of a transcription factor might be involved in this transition. According to our data, the presence of Rrn3 alone is not sufficient to trigger formation of Pol I monomers. However, it is possible that Rrn3 in addition to a yet unknown activity is required for dimer dissociation. Binding of Rrn3 might stabilize Pol I monomers resulting in a salt resistant initiation competent Pol I-Rrn3 complex[44]. At which stage of complex formation Rrn3 stabilizes the monomeric form remains to be unravelled. Future experiments are also necessary to elucidate the trigger and the mechanism for monomerisation and Pol I activation as well as the physiological relevance of Pol I dimers.

## Methods

**Yeast strains, plasmids, oligonucleotides and construction of transcription templates.** Yeast strains, plasmids and oligonucleotides used in this work are listed in Supplementary Tables 1, 2 and 3. Molecular biological methods and transformation of yeast cells were performed according to standard protocols[45–47]. The generation of transcription templates is described in the supporting information.

**Purification of transcription factors and Pol I subunits.** All factors were expressed in *E. coli* and purified according to protocols published in ref. 22 for Rrn3 and (refs 23,48) for CF and ref. 33 with some modifications.

**Purification of Rrn3.** Rrn3 was expressed as an N-terminal His-tag fusion protein in *E. coli* and purified according to ref. 22 with some modifications. Cells were lysed by sonication. The cleared lysate was loaded on Talon affinity resin (Clontech) equilibrated with Rrn3 lysis-buffer (50 mM HEPES/KOH; 10% glycerol; 200 mM KCl; 5 mM MgAc$_2$; 5 mM β-mercaptoethanol; and 5 mM imidazole). Beads were washed with Rrn3 lysis-buffer and protein was eluted with Rrn3 elution buffer (50 mM HEPES/KOH; 10% glycerol; 200 mM KCl; 5 mM MgAc$_2$; 5 mM β-mercaptoethanole; and 150 mM imidazole). Eluate was loaded onto an anion-exchange column (MonoQ 5/50 GL; GE heathcare) and eluated with an linear gradient from 20 to 80% buffer B; buffer A: (20 mM HEPES/KOH; 10% glycerol; 1 mM MgAc$_2$; 5 mM dithiothreitol (DTT)), buffer B: (20 mM HEPES/KOH; 10% glycerol; 1 mM MgAc$_2$; 5 mM DTT; and 1 M KCl). Each sample taken during the purification process was analysed via SDS–polyacrylamide gel electrophoresis (SDS–PAGE) to monitor the purification success and the protein concentration in the elution fraction was determined.

**Purification of CF.** CF subunits were coexpressed in *E. coli* and purified based on the procedures published by refs 23,48. Expression vector was a kind gift from B. Knutson. In brief, Recombinant CF protein was expressed by autoinduction in TB medium (1.2% tryptone; 2.4% yeast extract; 0.5% glycerol; 1/10 volume of a sterile solution containing 0.17 M KH$_2$PO$_4$ and 0.72 M K$_2$HPO$_4$ and 1/50 volume of a sterile solution containing 25% glycerol; 10% lactose and 2.5% glucose were added.) as described in ref. 49. A culture was grown at 37 °C to an OD600 of ∼0.6, after cooling the culture on ice, incubation was continued at 25 °C overnight. Cells were harvested (6,000 g; 10 min), resuspended in CF lysis buffer (50 mM HEPES/KOH; 10% glycerol; 5 mM MgAc$_2$; 500 mM KCl; 10 mM imidazole; 5 mM β-mercaptoethanole; 1 mM phenylmethylsulphonyl fluoride (PMSF); 2 mM benzamidine), treated with lysozyme (0,2 mg ml$^{-1}$; 30 min 4 °C on a spinning wheel) and lysed by sonication (Branson Sonifier 250 macrotip, 10 10 s pulses with 30 s cooling in icewater). The cell extract was cleared (40,000 g for 30 min at 4 °C) and incubated with 1 ml equilibrated NiNTA Agarose (Qiagen) at 4 °C for 2 h on a rotating wheel. The resin war transferred to a polypropylene column (Bio-Rad), washed with CF wash buffer 1 (20 mM HEPES/KOH; 10% glycerol; 5 mM MgAc$_2$; 1 M KCl; 20 mM imidazole; 5 mM β-mercaptoethanole; 1 mM PMSF; and 2 mM benzamidine), then CF wash buffer 2 (same as wash buffer 1 but 0.2 M KCl) and finally eluted with CF elution buffer (20 mM HEPES/KOH; 10% glycerol; 5 mM MgAc$_2$; 0,2 M KCl; 250 mM imidazole; and 5 mM β-mercaptoethanole). The eluate was loaded onto a MonoQ column (MonoQ GL 5/50 GE healthcare) a linear gradient from 20% buffer B to 80% buffer B was applied; buffer A: (20 mM HEPES/KOH; 10% glycerol; 1 mM MgAc$_2$; and 5 mM DTT), buffer B: (20 mM HEPES/KOH; 10% glycerol; 1 mM MgAc$_2$; 5 mM DTT; and 1 M KCl). The peak eluate fraction is loaded to a Superose6 HR 10/30 column (GE healthcare) equilibrated with buffer C: (20 mM HEPES/KOH; 10% glycerol; 1 mM MgAc$_2$; 5 mM DTT; and 0,15 M KCl).

**Purification of recombinant A49/34.5 complexes.** Expression and purification of recombinant A49/34.5 complexes and purification was according to ref. 33 with some modifications. Expression vector for the heterodimeric A49/A34.5 complex was a kind gift of P. Cramer and colleagues. Full-length A49/A34.5 or truncated A49 (amino acids 1–110 or 1–186) proteins were expressed by autoinduction in TB (ref. 49) overnight at 18 °C, harvested and resuspended in lysis buffer (50 mM

HEPES/KOH; 10% glycerol; 5 mM MgAc$_2$; 200 mM KCl; 10 mM imidazole; 5 mM β-mercaptoethanole; 1 mM PMSF; and 2 mM benzamidine), lysed by sonication and cleared by ultracentrifugation. The cleared lysate was incubated with 1 ml equilibrated Talon affinity resin (Clontech) at 4 °C on a rotating wheel for 2 h. The resin was washed in a polypropylene column (Bio-Rad), with A49 wash buffer 1 (20 mM HEPES/KOH; 10% glycerol; 5 mM MgAc$_2$; 0.2 M KCl; 10 mM imidazole; 5 mM β-mercaptoethanole; 1 mM PMSF; and 2 mM benzamidine) and finally eluted with elution buffer (20 mM HEPES/KOH; 10% glycerol; 5 mM MgAc$_2$; 0.2 M KCl; 150 mM imidazole; and 5 mM β-mercaptoethanole). The eluate was diluted with the same volume of buffer A (20 mM HEPES/KOH; 10% glycerol; 1 mM MgAc$_2$; and 3 mM DTT), loaded onto a MonoS column (MonoS GL 5/50; GE healthcare or MonoS HR 5/5 Amersham/Pharmacia) and eluted with a linear gradient of 10%—80% buffer B (20 mM HEPES/KOH; 10% glycerol; 1 mM MgAc$_2$; 3 mM DTT; 1 M KCl). The C-terminal domain of A49 (amino acids 186–415; or 111–415) was purified using the Talon affinity resin as described, but the anion-exchange chromatography step was omitted.

**Purification of yeast RNA polymerase I.** WT RNA Pol I and mutant polymerases were purified from yeast strains y2423 (Pol I WT) and y2670 (Pol I Δ49) according to ref. 34 with some modifications using a protein A (ProtA) affinity tag. The second largest subunit A135 is expressed as a C-terminal fusion protein with a protein A tag.

A 20 l YPD (yeast extract, peptone, dextrose) culture was grown at 30 °C to an optical density at 600 nm (OD600) of 2–3. Cells were harvested (4,000g for 6 min at room temperature), washed with ice-cold water, resuspended in buffer 1 (0,15 M HEPES/KOH pH 7.8, 40% glycerol, 60 mM MgCl$_2$, 3 mM DTT including protease inhibitors (PI; 2 mM Benzamidine, 1 mM PMSF) and adjusted to a final concentration of 400 mM (NH$_4$)2SO$_4$. Cell suspension was frozen in liquid nitrogen and stored at −80 °C. Subsequent steps were performed at 4 °C. Thawed cells were broken with glass beads using 10 cycles of bead beating. Glass beads and cellular debris was removed by centrifugation (4,000g for 20 min at 4 °C). The lysate was clarified by ultracentrifugation (100,000g for 90 min; Ti45 rotor Beckman Coulter).

Alternatively 2 l YPD (yeast extract, peptone, dextrose) culture was grown at 30 °C to an optical density at 600nm (OD600) of 1. Cells were harvested (4,000g for 6 min at room temperature), washed with ice-cold water, frozen in liquid nitrogen and stored at −20 °C. Cell pellet was weighed and resuspended in 1.5 ml lysis buffer (50 mM HEPES/KOH pH 7.8, 20% glycerol, 0.4 M (NH$_4$)$_2$SO$_4$, 40 mM MgCl2, 3 mM DTT including protease inhibitors (PIs (2 mM Benzamidine, 1 mM PMSF). Polymerase purification was performed at 4 °C. Cell suspension (0.7 ml) were added to 2 ml reaction tubes containing 1.4 g glass beads (diameter 0.75–1 mm, Roth). Cells were lysed on an IKA Vibrax VXR basic shaker with 2,200 r.p.m. at 4 °C for 10 min, followed by 5 min cooling on ice. This procedure was repeated four times. The cell extract was cleared from glass beads by perforation of the cup at bottom and cap and a centrifugation step (150g, 1 min, 4 °C). Cell debris were removed by centrifugation at 16,000g and 4 °C for 30 min. The protein content of the supernatant was determined using the Bradford assay.

Equal protein amounts (usually 1 ml cell extract, 20–30 mg) were incubated with 50–75 µl of immunoglobulin-G (rabbit serum, I5006-100MG, Sigma) coupled magnetic beads slurry (1 mm BcMag, FC-102, Bioclone)[50] for 1–2 h on a rotating wheel. The beads had previously been equilibrated three times with 500 µl lysis buffer. The beads were washed five times with 1 ml buffer B1500 (20 mM HEPES/KOH pH 7.8, 1 mM MgCl$_2$, 20% glycerol, 0.1% NP40, 1x PI and 1,500 mM KOAc) three times with 1 ml buffer B200 (same as B1500 but without NP40 and PIs and with 200 mM KOAc) and for elution finally resuspended in 50–100 µl buffer B200, supplemented with 3 µl Tobacco-Etch-Virus (TEV) protease (2,6 mg ml$^{-1}$) and incubated for 2 h at 16 °C, or overnight at 4 °C, in a thermomixer (1,000 r.p.m.). The supernatant was collected, the beads were washed twice with 50 µl B200 and the wash steps were added to the eluted fraction. Aliquots were frozen in liquid and stored at −80 °C. 10% of the elution fraction were analysed via SDS–PAGE to monitor the purification success. Protein concentration was determined by comparing the intensity of Coomassie-stained RNA polymerase subunits to the defined amounts of Coomassie-stained bovine serum albumin.

The same strategy was applied to strain y2670 from which Pol I depleted from subunits A49/34.5 (Pol I Δ49) was purified. Before cells were harvested an overnight preculture of strain y2670 was grown in Yeast extract-Peptone-Galactose (YPG) and then shifted for 24 h to glucose containing (YPD) medium to deplete subunit A49. Depletion of subunits A49 and A34.5 was monitored by Coomassie-stained SDS–PAGE and western blotting (Supplementary Fig. 6a).

The initiation competent Pol I-Rrn3 complex was purified from yeast whole-cell extracts according to ref. 34 with some modifications[51].

The same strategy was applied to strain yJPF162-1a (y2670) from which Pol I lacking subunits A49/34.5 (Pol I Δ49) was purified. Strain y2670 was cultivated in YPG and then shifted for 16 h to glucose containing (YPD) medium to deplete subunit A49 and A34.5. Depletion of subunits A49 and A34.5 was monitored by Coomassie-stained SDS–PAGE and western blotting (Supplementary Fig. 1).

**Purification of Pol I-Rrn3 complex.** The initiation competent Pol I-Rrn3 complex was purified from whole-cell extracts of strain y2183 which was transformed with plasmid YCplac111-GAL-Rrn3-ProtA (plasmid number 729). This strain overexpresses Rrn3- TEV-ProtA-His7 under the control of an GAL1/10 promoter. Purification was performed according to ref. 51 with some modifications. A 20-l fermentor was inoculated to an OD$_{600}$ ∼ 0.05 and the strain was grown overnight (doubling time ∼ 250 min) in YPR medium until the culture reached an OD$_{600}$ of 1–2. Rrn3 overexpression was induced for 3 h adding 2% galactose. The cells were harvested by centrifugation at 2,000g for 2 min. Cells were washed twice with ice-cold water and resuspended in yeast lysis buffer including 1 × PIs (1 ml buffer per 1 g of cell paste). The cell suspension was frozen in liquid nitrogen and stored at −80 °C. Cells were thawed on ice. All subsequent steps were performed at 4 °C. Cell suspension (250–300 ml) were placed in a 400-ml stainless steel bead beating chamber (Biospec) and glass beads (0.5 mm diameter) were added to fill the chamber. Cells were disrupted using 10 cycles of bead beating for 30 s and cooling for 90 s. The temperature of the surrounding ice/salt bath was kept at about −5 to −10 °C. Glass beads and cell debris were removed by centrifugation for 10 min at 4,000g. The supernatant was centrifuged at 100,000g for 90 min. The clear supernatant was carefully removed (∼300 ml). The clear supernatant was dialysed against buffer A (20% glycerol, 20 mM HEPES/KOH pH 7.8, 10 mM MgCl$_2$, 0.2 mM EDTA, 1 mM DTT, 1 × PIs) to reach the conductivity of buffer A90 (buffer A including 90 mM KCl). The dialysed sample was loaded on a DEAE-Sepharose column (13 × 5.2 cm), equilibrated with buffer A90, washed with 1 l buffer A90 and proteins were eluted with buffer A350 (buffer A including 90 mM KCl). The protein-containing peak fractions (between 200 and 300 ml) were dialysed against buffer B containing 1 × PIs overnight. The dialysed sample was centrifuged at 40,000g for 30 min at 4 °C. The pellet which contained the initiation competent Pol I was resuspended in 2 ml buffer B600 and the protein concentration was adjusted to 2.5–5 mg ml$^{-1}$. After centrifugation at 40,000g for 10 min to remove insoluble protein aggregates, equal protein amounts (10–20 mg) were incubated with 200 µl of immunoglobulin-G-coupled magnetic beads (1:1 slurry bead volume (100 µl); equilibrated three times with 500 µl buffer B600) for 2 h on a rotating wheel. The beads were washed four times with 1 ml buffer B1500 and then three times with 1 ml buffer B200 without NP40 and PIs. Then, the beads were resuspended in 100 µl buffer B200 supplemented with 11.7 µg TEV protease, and incubated for 2 h at 16 °C under shaking with 800 r.p.m. in a thermomixer. After collection of the supernatants, the beads were washed with 2 × 50 µl buffer B200 and both wash steps were added to the eluted fraction. 5–10% of the elution fraction were analysed in an SDS–PAGE. The protein concentration was adjusted to 0.05–0.1 pmol µl$^{-1}$.

**In vitro trancription.** *In vitro* transcription using tailed templates were executed as described in ref. 34 with the exception that no preincubation was performed. 25 nM of tailed templates were used in a total volume of 25 µl. Promoter-dependent *in vitro* transcription reactions were performed according to refs 51,52 with some modifications. In brief, 1.5 ml reaction tubes (Sarstedt safety seal) were placed on ice. 0.5–1 µl template (50–100 ng DNA) was added, which corresponds to a final concentration of 5–10 nM per transcription reaction (25 µl reaction volume). 1–2 µl CF (0.5 to 1 pmol µl$^{-1}$; final concentration 20–40 nM) and 1–3 µl Pol I-Rrn3 (final concentration 4–12 nM) were added to each tube. 20 mM HEPES/KOH pH 7.8 were added to a final volume of 12.5 µl. Transcription was started adding 12.5 µl transcription buffer 2 ×. The samples were incubated at 24 °C for 30 min at 400 r.p.m. in a thermomixer. 200 µl Proteinase K buffer (0.5 mg ml$^{-1}$ Proteinas K in 0.3 M NaCl, 10 mM Tris/HCl pH 7.5, 5 mM EDTA and 0.6% SDS) was added to the supernatant to stop transcription. The samples were incubated at 30 °C for 15 min at 400 r.p.m. in a thermomixer. Ethanol (700 µl) p.a. were added and mixed. Nucleic acids were precipitated at −20 °C overnight or for 30 min at −80 °C. The samples were centrifuged for 10 min at 12,000g and the supernatant was removed. The precipitate was washed with 0.15 ml 70% ethanol. After centrifugation, the supernatant was removed and the pellets were dried at 95 °C for 2 min. RNA in the pellet was dissolved in 12 µl 80% formamide, 0.1 M TRIS-Borate-EDTA (TBE), 0.02% bromophenol blue and 0.02% xylene cyanol. Samples were heated for 2 min under vigorous shaking at 95 °C and briefly centrifuged. After loading on a 6% polyacrylamide gel containing 7 M urea and 1 × TBE RNAs were separated applying 25 watts for 30–40 min. The gel was dried after 10 min rinsing in water for 30 min at 80 °C using a vacuum dryer. Radiolabelled transcripts are visualised using a PhosphoImager. For quantification, signal intensities were calculated using Multi Gauge (Fuji).

**Cryo-electron microscopy and Image Processing.** *Specimen preparation.* The purified Pol I complexes were cross-linked with 0.5% of glutaraldehyde for 2 min. and diluted to a final concentration of 7 µG ml$^{-1}$ in a buffer containing 20 mM Hepes, pH 7.8, 100 mM ammonium acetate and 2 mM MgCl$_2$. The specimen was adsorbed for 60 min. on a small piece of carbon partially floated off a mica sheet at the surface of a Teflon well containing 37 µl of sample. The carbon film was the transferred on an EM copper grid covered by a perforated carbon film (Quantifoil R2/2). The grid is flash frozen in liquid ethane using an automated plunger (Vitrobot, FEI) with controlled blotting time (4 s.), blotting force (5), humidity (95%) and temperature (20 °C). The particles were imaged using a cryo-transmission electron microscope (Titan Krios, FEI) equipped with a field emission gun operating at 300 kV and a Cs corrector. Images were recorded under low-dose condition (total dose of 22 e$^-$ Å$^{-2}$) using the automated data collection software EPU (FEI) on a 4,096 × 4,096 direct detector camera (Falcon II, FEI) at a magnification of × 59,000 resulting in a pixel size on the specimen of 0.108 nm. A total

of 17 frames with an electron dose of $3.2\,e^-\,Å^{-2}$/frame were collected and the frames 2–8 were aligned using the optical flow protocol[53] and averaged for further analysis. A high dose image ($55\,e^-\,Å^{-2}$), generated by summing all frames, was used as reference for frame alignment and for particle picking.

*Image processing.* Particles were selected manually from a subset of images using the boxer application in the EMAN2 software package (http://blake.bcm.tmc.edu/EMAN2/)[54]. The contrast transfer function of the microscope was determined for each micrograph using the CTFFIND3 programme[55] within the frame of the RELION software package[56], and the image phases were flipped accordingly. The molecular images were subjected to reference-free classification in RELION to produce representative two-dimensional (2D) class averages that were used as cross-correlation references for automated particle picking in all frames with the gEMpicker software[57]. The full data set was then partitioned in 2D using RELION to eliminate images containing contamination or bad particles. Structure refinement was done in RELION using a starting model obtained from a previously determined 3D model of Pol I derived from 2D crystals[58]. Three-dimensional reconstruction, structure refinement, 3D clustering and post-processing were carried out in RELION. The pol I dimer structure was first refined without imposing any symmetry constrains. The twofold symmetry axis was clearly identified on fitting the atomic structures of the monomers and was oriented in the z-direction before performing a 3D refinement with symmetry imposed. The initial rigid body fitting of the atomic structure of Pol I into the cryo-EM map was performed using gEMfitter[59]. Normal mode-based flexible fitting of the atomic structure of Pol I into the cryo-EM structures was performed using iMODFIT (ref. 60). Illustrations were prepared using the Chimera visualization software[61]. Molecular images of the Pol I-Rrn3 complex were extracted from 4,121 frames, while the WT Pol I molecule were selected from 2,934 frames.

*Local resolution estimation.* Local resolution estimation was performed using the 'ResMap' programme[62]. The data set was split randomly into two halves to obtain two reconstructions that were used to estimate the local resolution. The maps were coloured based on the local resolution estimation using the 'Surface color' tool implemented in Chimera.

**Data availability.** The cryo-EM maps have been deposited in the 3D-EM database (EMBL-European Bioinformatics Institute, Cambridge, UK). The EMBD accession code is EMD-14485 (Pol I–Rrn3).

The authors declare that the data supporting the findings of this study are available within the article and its Supplementary Information files.

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

## Acknowledgements

We are grateful to Drs Cramer, Hahn and Knutson for providing us with plasmids. We thank Dr Perez-Fernandez for constructing and providing us with strain y2670 and Dr Van Thai Huang for help with flexible fitting programs. This work was supported through grants of the Deutsche Forschungsgemeinschaft (SFB 960), the Fondation pour la Recherche Médicale, the Institut National de la Santé et de la Recherche Médicale, the Centre National pour la Recherche Scientifique, the Strasbourg University, the Association pour la Recherche sur le Cancer, the Agence Nationale pour la Recherche, the Labex INRT, the French Infrastructure for Integrated Structural Biology (FRISBI) [ANR-10-INSB-05-01] and INSTRUCT as part of the European Strategy Forum on Research Infrastructures (ESFRI).

## Author contributions

M.P. and R.S., performed biochemical experiments; G.P., C.C., F.K. and P.S. collected and analysed the cryo-EM data, and calculated and refined the EM densities. J.G. and P.M. participated in planning experiments, data interpretation and manuscript writing. P.S. and H.T. planned experiments and supervised the work. M.P., H.T. and P.S. prepared the figures and wrote the manuscript together.

## Additional information

**Competing financial interests:** The authors declare no competing financial interests.

