## [Peer Review File · Nature Communications]

REVIEWERS' COMMENTS:

Reviewer #1 (Remarks to the Author):

The authors have addressed all of the comments raised in the previous round of review.

Reviewer #2 (Remarks to the Author):

The points raised have been addressed

Reviewer 1

1. Is the resolution of these models sufficient to make clear conclusions when density is not observed (related to #6 below)?

Resolution and presence of density are two different issues. The presence of a density is related to the occupancy of a volume unit by an electron diffusion element, and its disappearance can be detected even though the secondary structural elements are not resolved. However we are in a favorable case since the spatial resolution of our cryo-EM maps is sufficient to distinguish alpha helices and to retrieve the global shape of beta sheets. Therefore when a density is missing, we can conclude which secondary structural element is missing.

However we agree with the referee that the criteria to consider that a density is missing is linked to the intensity threshold applied on the cryo-EM map. To decide whether a protein density was affected we first set an intensity threshold for which neighboring secondary elements are visible and at this value, the detail of interest has to be below the threshold. Furthermore, in a control experiment (the RNA polymerase I dimer), the detail of interest has to be above the threshold and be detected. (see discussion of point #6).

2. A short discussion of the recently published study on A12.2 by the Grill lab would be a nice addition.

A short discussion is now added on page 16. Consistently, Pol I recovers mostly by RNA cleavage for backtracks larger than 3 nt, whereas Pol II without TFIIS uses also 1D diffusion to regain transcription elongation. The “built in” cleavage activity may prevent frequently occurring transcriptional arrests which would be especially disadvantageous for the highly transcribed rRNA genes of fast dividing cells⁴².

3. Figure 1B: The epitope tags on the Rrn3 constructs are in different positions (though two recombinant Rrn3 expression plasmids are listed). Thus, this comparison of activity may not be biologically significant. The authors have previously shown that epitopes on the N- and C-termini affect the Rrn3 protein.

It was shown that epitopes at the N- and C-terminus of Rrn3 influence the stability of Rrn3 in yeast cells under distinct physiological conditions. Furthermore, different epitope tags can influence the solubility of recombinant Rrn3. To exclude that the position of epitope tags affects the activity of purified (soluble) Rrn3, we compared purified Rrn3 fractions containing different epitope tags in promoter-dependent in

vitro reactions. No significant differences in Rrn3 activity were observed. We included this information in Fig. S2. The transcription assay depicted in Fig. 1b was performed with His6-Rrn3. This is now mentioned in the Figure legend.

4. Furthermore, recombinant Rrn3 should be titrated against a constant Pol I amount. The use of an "optimized" amount is not clearly defined. Finally, the occupancy of Rrn3 on the polymerase should be tested in both cases.

A titration of recombinant Rrn3 against a constant Pol I amount is now presented in Fig. S2. This titration was in fact the basis for the definition of the "optimized" amount. We have modified the Results section accordingly (page 6).

*The "occupancy of Rrn3 on the polymerase" has only been tested thoroughly for the Pol I-Rrn3 complex. As stated in the Results section, cryo-EM suggested, that more than 3/5th of the polymerases are in complex with Rrn3 in the fraction containing the purified Pol I-Rrn3 complex (32.438 of 49.583 molecules) (see page 7). The result of negative staining EM performed with free Pol I incubated with over stoichiometric amounts of Rrn3 showed instead no significant increase in the ratio between monomeric and dimeric Pol I molecules (page 13, suppl. table 4). Although, the actual amount of Pol I-Rrn3 complexes was not analyzed in this experiment, this result suggested that Pol I-Rrn3 complex formation using purified components *in vitro* is poor under the conditions used. We have modified the Results and Discussion sections accordingly (page 6/7, page 13, suppl table 4, page 18).*

5. To argue against the model that Rrn3 influences the equilibrium between monomeric and dimeric states, the authors should show the data described (incubation with 10 fold excess Rrn3 for an extended time).

This is now included in suppl. table 4.

6. Can the EM data presented exclude occupancy of the active center by A12.2 C-terminal domain? The discussion section asserts that A12.2 is not an integral part of the active center. Is A12 possibly mobile? What is seen when the threshold is lowered, as done for A49/A34?

The density of A12 was investigated by setting different intensity threshold values on the cryo-EM map of the Rrn3-Pol I complex (see figure 1). At a value of 0.14 all the secondary elements (alpha helices, beta sheets) are above threshold, including the N-terminal part of A12.2. However the NTP entry pore where

the C-terminal part of A12.2 is located in the crystal structure has no density above threshold. At 0.08 some density appears above threshold in the pore but the most dense part of A12 (the two beta sheets that penetrate into the active site) are not revealed. Even at 0.04 and 0.02 these densities are not resolved while noise starts to appear above threshold in the whole map. The density which appears in the pore does not reflect the density of A12 and is most probably noise. These results indicate that the A12 site is very weakly occupied in the Rrn3-Pol I complex since the pore fills up only at a very low threshold and since the shape is not compatible with A12.

Figure 1 Variation of the intensity threshold in the cryo-EM map of the Pol I-Rrn3 complex

Conversely we performed the same experiment with the Pol I dimers (figure 2). At a value of 0.14 all the secondary elements (alpha helices, beta sheets) are above threshold. No density for the C-terminal part of A12.2 can be detected in the NTP entry pore and the A49 density is also missing. At 0.10 and more clearly at 0.08 the shape of the C-terminal part of A12.2 is fully recognized. At these thresholds the signal is also seen for A49. Lower threshold values mainly reveal noise

These results indicate that the C-terminus of A12 and the A49 sites are partially occupied in the Pol I dimer. However the shapes of the A12 and the A49 secondary clearly appear at a lower threshold, thus identifying the densities.

We included a statement about the possible mobility of A12 in the discussion section (page 14), and added the analysis lowering the A12 threshold to the rebuttal letter and as a supplemental figure 7.

Figure 2 Variation of the intensity threshold in the cryo-EM map of the Pol I dimer complex

Reviewer 2

Data & Methodology

- **One or a few representative micrographs along with 2D class averages should be shown. Fourier Shell Correlation curves should also be shown.**

Original images, class averages as well as Fourier Shell Correlation functions are now shown for each data set in the supplemental figures 3 and 6.

- **Details about the reconstruction of the Pol I dimer should be provided (symmetry used ...).**

A more detailed description of the protocol used for dimer refinement is given on page 19.

- **It should be stated early on in the main text of the manuscript the sample has been crosslinked.**

The fact that the sample has been cross-linking is now stated in the results section on page 6.

- **In Fig 1b, the graphs for initiation activity and non-specific activity should be presented in the same order for clarity.**

Graphs are now presented in the same order.

- **The normal-mode analysis used to obtain pseudo-atomic models is not described in the Methods section. Also, the authors should justify the use of flexible fitting vs rigid-body fitting at this resolution.**

The initial rigid body fitting of the atomic structure of the RNA polymerase into the cryo-EM map was done using gEM-fitter (gEMfitter: A highly parallel FFT-based 3D density fitting tool with GPU texture memory acceleration", T. Van Hoang, X. Cavin, and D.W. Ritchie, Journal of Structural Biology, 2013, 184, 348-354.) a program for multiresolution fitting of macromolecular structures. Its main purpose is to fit high resolution protein structures into a low resolution cryo-EM density map.

The initial fitting clearly showed that large parts of the structure had undergone significant concerted movements. In particular long helices were clearly out of register. We therefore used Pablo Chacón's iMODFIT program that allows flexible fitting of atomic structures into EM Maps based on uses Normal Mode Analysis in internal coordinates. (iMODFIT: efficient and robust flexible fitting based on vibrational analysis in internal coordinates (2013). López-Blanco J.R. and Chacón P. JSB 184(2):261–27.) These references are now cited in the methods section on page 19.

References

- **A reference is missing for the optical flow protocol used for aligning movie frames.**

We thank the referee for having noticed this oversight. The following reference was added in the text. (Abrishami V, Vargas J, Li X, Cheng Y, Marabini R, Sorzano CO, Carazo JM. Alignment of direct detection device micrographs using a robust Optical Flow approach. J Struct Biol. 2015

Mar;189(3):163-76.)

Clarity and context

- **For clarity, the authors should consider adding a figure showing the crosslinks reported in reference 22 as they are frequently referred to in the text.**

We made a supplemental figure S4 to show the crosslinked residues and the distances in the model.

Reviewers' Comments:

Reviewer #1 (Remarks to the Author)

Review of Pilsl, et al. Nature Communications 2016

General Comments

In this manuscript, the authors present three cryo-EM reconstructions of RNA polymerase I. These three models represent three unique states of the enzyme in solution: bound to Rrn3, monomer, and dimer. They then compare the features of these models to the published, high resolution model for dimeric Pol I. They find that most of the critical differences observed between Pol II and Pol I crystal structures can be explained by the fact that Pol I was a dimer in the crystal structures. Previous genetic and biochemical observations for the role of Rrn3 in transcription initiation were confirmed by the new cryo-EM structure.

The approaches employed in this study are carefully controlled and described. The biochemical assays provided are descriptive, but largely supportive of the assertions based on the structural data. Though the overall study is largely confirmatory of previous work, there are important distinctions made here, compared to the previously published structures from the Cramer and Mueller labs. The use of cryo-EM enables the authors to test the role for dimerization in the adoption of the "expanded" structure. Other than the partially unfolded bridge helix, the cryo-EM model is quite similar to the expected structure for an elongation-competent, multisubunit RNA polymerase.

Specific Comments

1. Is the resolution of these models sufficient to make clear conclusions when density is not observed (related to #5 below)?
2. A short discussion of the recently published study on A12.2 by the Grill lab would be a nice addition.
3. Figure 1B: The epitope tags on the Rrn3 constructs are in different positions (though two recombinant Rrn3 expression plasmids are listed). Thus, this comparison of activity may not be biologically significant. The authors have previously shown that epitopes on the N- and C-termini affect the Rrn3 protein.
4. Furthermore, recombinant Rrn3 should be titrated against a constant Pol I amount. The use of an "optimized" amount is not clearly defined. Finally, the occupancy of Rrn3 on the polymerase should be tested in both cases.
5. To argue against the model that Rrn3 influences the equilibrium between monomeric and dimeric states, the authors should show the data described (incubation with 10 fold excess Rrn3 for an extended time).
6. Can the EM data presented exclude occupancy of the active center by A12.2 C-terminal domain? The discussion section asserts that A12.2 is not an integral part of the active center. Is A12 possibly mobile? What is seen when the threshold is lowered, as done for A49/A34?

Reviewer #2 (Remarks to the Author)

Summary of the key results.

Pilsl et al report the cryoEM characterization of the yeast Pol I-Rrn3 complex and of the free Pol I monomer and dimer. All reported reconstructions have subnanometer resolutions and allowed the authors to fit known crystal structures of Pol I and Rrn3. Rrn3 is observed to bind to Pol I via interactions with A43/A14, A190 and AC40, in agreement with the literature. The authors also report that Rrn3 binding to Pol I is incompatible with Pol I dimerization leading them to put forward that Rrn3

stabilizes a specific monomeric initiation competent form of Pol I and drives preinitiation complex formation.

I recommend publication of this manuscript provided the points raised below are addressed.

Originality and interest

This is the first structure of PolI in complex with Rrn3 and constitute a step forward in the understanding of rRNA transcription in eukaryotes.

Data & Methodology

One or a few representative micrographs along with 2D class averages should be shown.

Fourier Shell Correlation curves should also be shown.

Details about the reconstruction of the Pol I dimer should be provided (symmetry used ...).

It should be stated early on in the main text of the manuscript the sample has been crosslinked.

In Fig 1b, the graphs for initiation activity and non-specific activity should be presented in the same order for clarity.

The normal-mode analysis used to obtain pseudo-atomic models is not described in the Methods section. Also, the authors should justify the use of flexible fitting vs rigid-body fitting at this resolution.

References

A reference is missing for the optical flow protocol used for aligning movie frames.

Clarity and context

For clarity, the authors should consider adding a figure showing the crosslinks reported in reference 22 as they are frequently referred to in the text.